# Determinants of Parental Intention to Vaccinate Young Adolescent Girls against the Human Papillomavirus in Taiwan: An Online Survey Study

**DOI:** 10.3390/vaccines12050529

**Published:** 2024-05-11

**Authors:** Pei-Yun Lin, Tai-Ling Liu, Li-Ming Chen, Meng-Jung Liu, Yu-Ping Chang, Ching-Shu Tsai, Cheng-Fang Yen

**Affiliations:** 1Department of Psychiatry, Kaohsiung Medical University Hospital, Kaohsiung Medical University, Kaohsiung 80754, Taiwan; 2Department of Psychiatry, School of Medicine, College of Medicine, Kaohsiung Medical University, Kaohsiung 80708, Taiwan; 3Institute of Education, College of Social Sciences, National Sun Yat-sen University, Kaohsiung 80404, Taiwan; 4Department of Special Education, College of Education, National Kaohsiung Normal University, Kaohsiung 80201, Taiwan; 5School of Nursing, The State University of New York, University at Buffalo, New York, NY 14214, USA; 6Department of Child and Adolescent Psychiatry, Chang Gung Memorial Hospital, Kaohsiung Medical Center, Kaohsiung 83301, Taiwan; 7School of Medicine, Chang Gung University, Taoyuan 33302, Taiwan; 8College of Professional Studies, National Pingtung University of Science and Technology, Pingtung 91201, Taiwan

**Keywords:** parent, girl, vaccine, human papillomavirus

## Abstract

Since 2018, Taiwan has included the human papillomavirus (HPV) vaccination into its national immunization program for junior high school girls. However, the reports of side effects following vaccination have increased parental concerns. This study investigated parental intentions regarding the HPV vaccination for their daughters and related factors in Taiwan. A total of 213 parents of girls aged between 12 and 15 years participated in an online survey. The survey collected data on various factors, including the parental intention to vaccinate their daughters against HPV; the motivation behind the vaccinations, as measured using the Motors of Human Papillomavirus Vaccination Acceptance Scale; an understanding of the reasons behind the government’s promotion of HPV vaccinations; concerns regarding the side effects of vaccinations for their daughters; an awareness of the reported side effects of HPV vaccines experienced by some individuals; the exposure to information on HPV vaccines from social media; and mental health status, measured using the Brief Symptom Rating Scale. The associations between these variables and the parental intention to vaccinate their daughters against HPV were examined using a multivariable linear regression analysis model. The findings revealed a moderate to high level of intention among participants to vaccinate their daughters against HPV. Parents who perceived a greater value in HPV vaccination for their daughters’ health (B = 0.524, standard error [se] = 0.039, *p* < 0.001) and had greater autonomy in decision-making regarding vaccination (B = 0.086, se = 0.038, *p* = 0.026) exhibited a higher intention to vaccinate their daughters against HPV. Conversely, parents who expressed greater concern regarding the side effects of HPV vaccines for their daughters had a lower intention to vaccinate (B = −0.762, se = 0.203, *p* < 0.001). Based on these findings, this study recommends integrating these factors into the design of intervention programs aimed at enhancing parental intentions to vaccinate their daughters against HPV.

## 1. Introduction

The human papillomavirus (HPV) is a DNA virus that targets the epidermis and mucous membranes of the body, with more than 200 identified types classified as either low-risk or high-risk [1]. Among the high-risk strains, HPV types 16 and 18 are particularly prevalent, and are associated with the development of cervical precancerous lesions and cervical carcinoma, accounting for more than 70% of cervical cancers [1]. Additionally, HPV infection has been demonstrated to increase the risks of oral malignancies [2]. Research indicates that HPV can promote tumorigenesis by targeting specific genes, proteins, and signaling pathways through its E6 and E7 oncoproteins, which inhibit two crucial tumor suppressors, P53 and Rb [3]. HPV types 6 and 11 are the most common low-risk strains and are known to cause genital warts [1]. HPV infection has also been identified as a risk factor for spontaneous abortions [4]. Furthermore, HPV infection is speculated to increase the risk of male infertility [4]. HPV is mainly transmitted through sexual contact involving skin, mucous membranes, or bodily fluids, including during sexual intercourse. Moreover, transmission can occur through contact with objects carrying HPV on external genitalia [1].

Both men and women carry a 50–80% risk of contracting HPV during their lifetime, with the risk drastically increasing for women upon the initiation of sexual activity [1]. A meta-analysis demonstrated the correlations of systemic immunosuppression (including HIV infection), the use of immunosuppressive medications for inflammatory bowel disease, and alterations in the vaginal microbiome with the persistence of HPV, the rate of cervical intra-epithelial neoplasia, and the rate of cervical cancer [5]. Additionally, factors such as smoking, a high number of sexual partners, and an early age at first pregnancy contribute to the increased risk of HPV infection [5]. Given its potentially catastrophic health effects, the prevention of HPV infection has become a global health-care priority.

Currently, the United States Food and Drug Administration (FDA) has approved three prophylactic vaccines for HPV [6]. Gardasil (Merck) was the first vaccine to be approved by the FDA in 2006 and provides protection against HPV genotypes 6, 11, 16, and 18. Next, Cervarix (GlaxoSmithKline) was approved by the FDA in 2009 and provides immunity against HPV16 and HPV18. The FDA further approved the supplemental use of Gardasil 9 (HPV nine-valent vaccine) in 2014 to expand its protection against HPV 6, 11, 16, 18, 31, 33, 45, 52, and 58 [7,8,9,10]. The design of the currently available vaccines is based on virus-like particles (VLPs) that are generated by the major capsid protein L1 and mimics the structure of virions [11,12,13,14,15]. Because VLPs do not contain viral genome, they are considered safer than attenuated or inactivated viruses that could be turned into infectious ones [8,9]. Vaccination against HPV generates prolonged, 10- to 100-fold higher titers of L1 specific neutralizing antibodies compared to natural infection [7,8,9,15]. However, the efficacy of HPV vaccines varies depending on age. A review of 21 published studies revealed that the highest effectiveness of HPV vaccines was observed in the youngest age bracket. Specifically, vaccine effectiveness estimates ranged from approximately 74% to 93% for adolescents aged 9–14 years and from 12% to 90% for those aged 15–18 years [16]. Therefore, the World Health Organization (WHO) recommends targeting females aged 9–14 years as the primary demographic for HPV vaccination. Moreover, achieving a vaccination rate of 80% or higher in this age group could mitigate the risk of HPV infection in males [17].

In accordance with WHO’s recommendation, Taiwan’s Health Promotion Administration has been actively promoting the free HPV vaccination service. The first step was to subsidize a free HPV vaccination for junior high school girls, which has been ongoing since 2018 [18]. The goal is to enhance the immunity of young Taiwanese females before they become sexually active, thereby reducing the risk of HPV infection [18]. Taipei City will also subsidize free Gardasil 9 vaccinations for junior high school boys from September 2024 onwards [19]. However, cases of juvenile rheumatoid arthritis have been reported in adolescent girls in Taiwan following the HPV vaccination [20]. Furthermore, recurring news coverage detailing a variety of symptoms experienced by some recipients of the vaccine in Japan, including chronic pain and motor impairment has instilled concern among parents in Taiwan [21].

In April 2013, the Japanese government introduced the HPV vaccine into the national immunization program for girls aged 12–16 years old. However, merely 2 months later, the proactive recommendation for its routine use was halted due to public concerns regarding the potential side effects [21]. The widespread coverage of the side effects of HPV vaccines in the media, coupled with the Japanese government’s suspension of HPV vaccination recommendations, negatively affected parental intention to vaccinate their daughters against HPV [22]. Despite subsequent clarifications from global health authorities, such as the WHO’s statement in 2014—which asserted that the reported adverse events in Japan were unrelated to the HPV vaccine—and the 2016 Global Committee on Vaccine Safety (GACVS) assessment—which revealed no scientific evidence linking serious adverse events to the HPV vaccine—reluctance toward the HPV vaccination persisted. Additionally, a synthesis of 26 clinical studies by the WHO in 2017 highlighted a lack of adverse reactions following HPV vaccine administration [21]. However, increasing the Japanese acceptance of HPV vaccination remains challenging. A 2022 study revealed that many Japanese university students possess inadequate knowledge regarding the HPV vaccine—related morbidity, mortality, and preventive measures [23]. Similarly, a study conducted in 2023, involving 3790 Japanese women, reported that 28.2% expressed an acceptance of the HPV vaccination, with 51.6% adopting a neutral stance and 20.1% exhibiting negative attitudes and a low intention to receive the HPV vaccination [24]. A higher level of knowledge and higher incomes were significantly associated with a higher vaccination intention, whereas concerns regarding adverse effects and an occupation as a worker or unemployed status were associated with lower vaccination intention [23,24].

In Taiwan, some civil organizations and politicians have raised doubts regarding the necessity of the government’s publicly funded vaccination subsidies for young girls [20]. Despite a study in Taiwan demonstrating no statistically significant increases in the risk of 19 selected serious adverse events and indicating no association between HPV vaccination and serious adverse events [25], concerns among parents regarding the potential side effects of HPV vaccines may still influence their intention to vaccinate their children. Given the various challenges related to the government’s policy to promote HPV vaccination for young girls, including resistance from civil organizations and politicians, parental intentions to vaccinate their daughters against HPV and the related factors in Taiwan warrant further study. The findings of this study will provide valuable insights for the government to refine its policy on promoting the HPV vaccination for adolescents and address parental concerns effectively.

Researchers have employed both the theory of planned behavior (TPB) [26] and protection motivation theory (PMT) [27] to investigate the factors influencing parental intention to vaccinate their daughters against HPV. According to the TPB, three main factors influence behavioral intention: attitude (an individual’s overall evaluation of whether the behavior is favorable or unfavorable), subjective norm (the perceived social pressure to engage in or refrain from the behavior), and perceived behavioral control (the individual’s belief in their ability to perform the behavior) [26,28]. The PMT posits that the perceived severity of the health threat caused by illnesses and vulnerability to illnesses (threat appraisals) and the perceived efficacy of coping behaviors in alleviating the threat of illnesses (coping appraisals) influences behavioral intention [29]. Moreover, the emergence of social media as a vital platform for disseminating HPV vaccine information has altered information processing patterns [30], despite the prevalence of health misinformation across the majority of social media platforms [31]. A study employing the PMT approach revealed that information sources affected parents’ perceptions of HPV severity and susceptibility and vaccine response efficacy [32]. Moreover, parents’ perceptions of HPV vaccines, rather than perceptions of HPV itself, influenced their intention to vaccinate their daughters [32]. Concerns regarding vaccine side effects also contribute to parental hesitancy toward HPV vaccines [33]. Moreover, parental mental health may influence their decisions regarding vaccination for their children [34]. However, a comprehensive examination of the factors influencing parental intentions to vaccinate their daughters against HPV has yet to be conducted on a nationwide scale in Taiwan.

The primary objective of this study was to investigate the associations of various factors with parental intention to vaccinate their daughters against HPV. Specifically, we hypothesized that parents have a greater intention to vaccinate their daughters against HPV when they have the following: a stronger perception of the value of the HPV vaccination for their daughters’ health; greater knowledge regarding vaccinations; greater autonomy in decision-making regarding vaccinations; a better understanding of the government’s rationale for promoting the HPV vaccination; less concerns regarding the side effects of the vaccination for their daughters; less awareness of the side effects of HPV vaccines experienced by some individuals; greater exposure to information on HPV vaccines from social media; and better mental health.

## 2. Methods

### 2.1. Participants and Procedure

This study enrolled parents residing in Taiwan who had daughters aged between 12 and 15 years old. Girls in this age bracket are likely to be in junior high school in Taiwan, and thus are likely to be eligible for free vaccination subsidized by Taiwan’s Health Promotion. Participants were recruited through online advertisements posted on social media platforms, such as Facebook, Twitter, LINE (a direct messaging app commonly used in Taiwan), and PTT (a popular online forum in Taiwan) from January 2024 to March 2024. The advertisement provided details about the study’s objectives, instructions on how to complete the questionnaire online, and assurances regarding the protection of respondents’ privacy. Interested parents were able to access the online questionnaire by clicking the “Agree to Participate” button and proceeding to provide their responses. Those who chose not to participate could opt out by clicking the “Not willing to participate” button or simply by disregarding the online advertisement.

A total of 261 parents responded to the advertisement. Among them, 14 parents indicated their unwillingness to participate by clicking the corresponding button. Additionally, 34 parents completed the online questionnaire but were subsequently excluded due to their daughters’ age falling outside the specified range of 12 to 15 years old. Consequently, the data of 213 parents were included in the analysis. According to Green [35], a total number of participants in the studies used a regression analysis model needs at least 50 + 8 * (the number of independent variables). There were 13 independent variables in this study; this study needed at least 154 participants. Therefore, 213 parents were enough for the regression analysis used in this study.

### 2.2. Ethics Statement

This study was approved by the institutional review board (IRB) of Kaohsiung Medical University Hospital (KMUHIRB-EXEMPT(I)-20240001). Considering the anonymous nature of the online questionnaire survey, the IRB waived written informed consent. The questionnaire-based study did not involve experiments on humans or human tissue samples. Furthermore, the study adhered to the principles outlined in the Declaration of Helsinki and the guidelines for the Conduct, Reporting, Editing, and Publication of Scholarly Work in Medical Journals.

### 2.3. Measures

#### 2.3.1. Parental Intention to Vaccinate Daughters against HPV

In this study, participants were asked about their intention to vaccinate their daughters against HPV by using the following question: “Please rate your current willingness to let your daughter receive HPV vaccination”. Responses were rated on a Likert scale ranging from 1 (*very low*) to 10 (*very high*) [36].

#### 2.3.2. Twelve-Item Parental Motors of the Human Papillomavirus Vaccination Acceptance Scale

The 12-item traditional Chinese version of the Parental Motors of Human Papillomavirus Vaccination Acceptance Scale (P-MoVac-HPV-12) was used to assess the participants’ motivation to vaccinate their daughters against HPV. The research team adapted the 12-item parent version of the Motors of COVID-19 Vaccination Acceptance Scale (P-MoVac-COVID19S-12) [37] into the P-MoVac-HPV-12 by substituting the term “COVID-19” with “human papillomavirus”. For example, the original item 7, “I feel pressured about letting my child receive the COVID-19 vaccine” was modified to “I feel pressured about letting my child receive the human papillomavirus vaccine”. Similarly, item 11, “I only let my child receive the COVID-19 vaccine if it is required” was altered to “I only let my child receive the human papillomavirus vaccine if it is required”. The P-MoVac-HPV-12 scale comprises four domains, each containing four items. These domains pertain to the respondent’s values (e.g., “Vaccinating my child against human papillomavirus is important”), the perception of the effects of vaccination (e.g., “Vaccination greatly reduces my child’s risk of human papillomavirus infection”), knowledge regarding vaccination (e.g., “I understand how the vaccine helps my child’s body fight the human papillomavirus”), and autonomy in decision-making regarding their child’s health care (e.g., “I can choose whether to allow my child to be vaccinated against human papillomavirus or not”). Participants rated each item on a 7-point Likert-type scale (1 = *strongly disagree*; 7 = *strongly agree*), with three items (i.e., items 7, 10, and 11) reverse-coded to ensure that higher total scores on the P-MoVac-HPV-12 reflected the greater levels of parental acceptance of vaccinating their children against human papillomavirus. The validity and reliability of the 12-item P-MoVac-COVID19S scale has been established [37]. In this study, the Cronbach’s α coefficients for the four domains of the P-MoVac-HPV-12 ranged from 0.820 to 0.902.

#### 2.3.3. Five-Item Brief Symptom Rating Scale

The 5-Item Brief Symptom Rating Scale (BSRS-5) was used to assess the mental health status of parents [38]. The scale comprises five items related to the various aspects of psychopathology: (1) feeling tense or keyed up (anxiety); (2) feeling low in mood (depression); (3) feeling easily annoyed or irritated (hostility); (4) feeling inferior to others (interpersonal hypersensitivity: inferiority); and (5) having trouble falling asleep (insomnia). The parents were instructed to rate the severity of these symptoms on a 5-point scale ranging from 0 (*not at all*) to 4 (*extremely*). A total BSRS-5 score of 6 or higher indicates a poor mental health status [38]. The BSRS-5 has been demonstrated to possess satisfactory psychometric properties, making it a reliable measure for detecting psychiatric morbidity both in medical settings and community contexts [38]. In this study, Cronbach’s α coefficient for the BSRS-5 was 0.894.

#### 2.3.4. Understanding of the Government’s Rationale for Promoting the HPV Vaccination

In this study, the participants’ level of understanding the government’s rationale for promoting HPV vaccination was assessed using the following question: “To what extent do you understand why the government advocates for HPV vaccination among the junior high school girls?” The participants rated their level of understanding on a 4-point scale ranging from 0 (*not at all*) to 3 (*realize very well*).

#### 2.3.5. Concerns Regarding the Side Effects of the HPV Vaccine

In this study, the participants level of concern regarding the potential side effects of HPV vaccines for their daughters was assessed using the following question: “How concerned are you about the possible side effects of HPV vaccine for your daughter?” The participants rated their level of concern on a 4-point scale ranging from 0 (*not at all*) to 3 (*extremely*).

#### 2.3.6. Awareness of the Reported Side Effects of the HPV Vaccination Experienced by Some Individuals

The participants were asked the following question: “Have you encountered anyone who experienced side effects following HPV vaccination?” They were given the option to respond with either “yes” or “no”.

#### 2.3.7. Exposure to Information about the HPV Vaccine from Social Media

The participants were asked the following question: “Have you ever obtained information about HPV vaccines from social media platforms such as Line and Facebook groups?” They were given the option to respond with either “yes” or “no”.

#### 2.3.8. Demographic Characteristics of the Parents and Children

Information on the sex (women vs. men), age, and education level (high school or below vs. college or above) of the parents as well as the age of their daughters were collected for analysis.

### 2.4. Statistical Analysis

Statistical analyses were conducted using IBM SPSS Statistics version 24.0 (IBM Corporation, Armonk, NY, USA). The continuous variables are presented as means (standard deviations [SD]) and the category variables are presented as frequencies (percentages). The normal distribution of continuous variables was assessed using the criteria of absolute values of <4 and <2 for kurtosis and skewness, respectively [39,40]. These tests indicated that the data were normally distributed. Bivariable linear regression analysis was employed to investigate the associations between parental factors, demographic characteristics, and parental intentions to vaccinate their daughters against HPV. Factors that were found to be significantly associated with parental intention were further analyzed in a stepwise multivariable linear regression model. A two-tailed *p* value of <0.05 indicated statistical significance.

## 3. Results

A total of 213 participants (172 women and 41 men) participated in this study. Table 1 presents the characteristics of the participants and their daughters. The average age of the participants was 46.2 years (SD = 4.1 years), whereas the mean age of their daughters was 13.8 years (SD = 1.0 years). More than three-fourths (76.1%) of the participants had attained an education degree of college or above, with 22.5% reporting a poor mental health status. The participants’ mean scores for the four domains of the P-MoVac-HPV-12 ranged between 13.9 (SD = 3.4) and 16.9 (SD = 3.6). Moreover, the participants’ mean score for the understanding of the government’s rationale for promoting HPV vaccination was 1.9 (SD = 0.8). Furthermore, the participants’ mean score for their concerns regarding the side effects of HPV vaccines for their daughters was 1.5 (SD = 0.7). Nearly two-fifths (39.9%) of the participants had heard of the instances of side effects after HPV vaccination, whereas 55.4% had received information about the HPV vaccine from social media. The mean score for parental intention to vaccinate their daughters against HPV was 7.4 (SD = 2.9).

Table 2 presents the results on the factors associated with parental intention to vaccinate their daughters against HPV, as determined through bivariable linear regression analysis. Fathers exhibited a higher intention to vaccinate their daughters against HPV compared to mothers (*p* = 0.049). Moreover, higher scores in the four domains of parental motivation to vaccinate their daughters against HPV, namely, vaccination effects, knowledge, values, and autonomy, were significantly associated with higher parental intention to vaccinate (all *p* < 0.001). Furthermore, a greater understanding of the government’s rationale for promoting HPV vaccination was significantly associated with a higher parental intention to vaccinate their daughters against HPV (*p* < 0.001). Conversely, a greater concern regarding the side effects of HPV vaccines for their daughters was significantly associated with lower parental intention (*p* < 0.001). Parents who were aware of reported side effects of HPV vaccines experienced by some individuals were more inclined to have a lower intention to vaccinate their daughters compared with those who had never encountered such information (*p* = 0.004). Notably, parents’ age, education level, mental health status, and exposure to information about HPV vaccines from social media and their daughters’ age were not significantly associated with parental intention to vaccinate daughters against HPV (all *p* > 0.05).

The factors significantly associated with the parental intention to vaccinate daughters against HPV in the bivariable linear regression models were subsequently included in a multivariable linear regression model (Table 3). The analysis revealed that parents’ perceived value of the HPV vaccination for their daughters’ health was the first factor that was entered into the model and was significantly associated with a higher parental intention to vaccinate their daughters against HPV (*p* < 0.001). Parents’ concerns regarding the potential side effects of the vaccination for their daughters was the second factor entered into the model and was significantly associated with a decreased parental intention to vaccinate their daughters against HPV (*p* = 0.026). Lastly, parents’ autonomy in deciding whether to have their daughters vaccinated against HPV was the third factor that was entered into the model. This factor was significantly associated with higher parental intention (*p* < 0.001).

## 4. Discussion

The study findings revealed that parents, on average, rated their intention to vaccinate their daughters against HPV at 7.4 on a 10-point Likert scale. Furthermore, a multivariable linear regression analysis model indicated that parents who more strongly perceived the value of the HPV vaccination for their daughters’ health and who had greater autonomy in decision-making regarding vaccinations exhibited a higher intention to vaccinate their daughters against HPV. Conversely, parents who expressed a greater concern regarding the potential side effects of vaccination for their daughters exhibited a lower intention to vaccinate them against HPV.

In this study, parents expressed a moderate to high level of intention to vaccinate their daughters against HPV, with a mean score of 7.4 on a 10-point Likert scale. However, following the introduction of publicly funded subsidized HPV vaccination for junior high school girls in Taiwan, concerns arose due to reports of generalized joint pain diagnosed as juvenile rheumatoid arthritis postvaccination. Moreover, before our survey, the dissemination of Japanese reports on the side effects of HPV vaccination by Taiwanese media have contributed to parental unease regarding their decision about whether to have their children vaccinated. Some civil organizations and politicians questioned the allocation of government funds for HPV vaccination, arguing that cervical cancer is no longer a leading cause of female deaths in Taiwan. Nonetheless, the government continued its efforts to emphasize the importance of the HPV vaccination for parents. The bivariable linear regression analysis revealed that a better understanding of the government’s rationale for promoting the HPV vaccination was significantly associated with higher parental intention, underscoring the necessity of actively communicating well-intentioned policies to the public to enhance acceptance. However, some parents remained hesitant to have their daughters vaccinated against HPV, as evidenced by 15 (7.0%) participants who rated their intention to vaccinate as *very low* (score of 1). Considering the benefits of HPV vaccination, educating parents about its importance and strengthening their intention to have their daughters vaccinated is essential.

This study highlighted that the perceived value of the HPV vaccination for their daughters’ health among parents was the most significant factor associated with parental intention to vaccinate their daughters against HPV. Both a favorable attitude toward vaccination in the TPB [26] and the perceived efficacy of vaccination in the PMT [27] were the core factors influencing individuals’ vaccination intentions. These findings align with previous research [30,32], underscoring the importance of using scientific evidence to educate parents about the efficacy of HPV vaccines in mitigating the risk of contracting HPV and developing associated diseases among adolescent girls in the future.

Parental concern regarding the side effects of vaccination for their daughters was determined to be the second most influential factor associated with the intention to vaccinate their daughters against HPV. Additionally, having heard of the side effects of HPV vaccines occurring in some individuals was linked to parents’ lower intention to vaccinate in the bivariable regression analysis model. According to the TPB [26], vaccine side effects can contribute to parents’ negative attitudes toward vaccination and reduce their intention to vaccinate their daughters. This finding aligns with that of a previous study [33]. Addressing parental concerns by providing information on the potential side effects of the HPV vaccine, on current incidence rates, and on how HPV vaccine side effects can be identified is crucial.

Parental autonomy in deciding whether to have their daughters vaccinated against HPV was positively correlated with their intention. Reports of serious side effects following the HPV vaccination in Taiwan and Japan have led to public skepticism regarding the safety of vaccinating their children, potentially affecting parental autonomy. Additionally, opposition to vaccines from other family members may make parents more hesitant in sending their children for a vaccination. This finding aligns with that of a previous study [28]. The study results underscore the importance of health-care professionals assisting parents in understanding the advantages and disadvantages of vaccinating their daughters and supporting their autonomy in making informed decisions and taking action.

Social media serves as a crucial platform for disseminating information about HPV vaccines [30]. Receiving information about HPV vaccines through social media not only enhances parents’ knowledge but also raises awareness of the subjective norm surrounding HPV vaccination. Therefore, exposure to information on HPV vaccination from social media was hypothesized to be positively correlated with the parental intention to vaccinate. However, the findings of this study did not support this hypothesis. Furthermore, poor mental health was hypothesized to impede parents’ cognitive function and reduce their ability to make decisions regarding the vaccination of their children [34]. However, the results of this study did not support this hypothesis.

This study provides the following recommendations. Health professionals should develop strategies for enhancing parental intention to vaccinate their daughters against HPV. Intervention programs should focus on educating parents about the effectiveness of HPV vaccines in mitigating the risk of HPV infection in adolescent girls, as well as the potential vaccine side effects. Strengthening parents’ confidence in the benefits of HPV vaccines and alleviating their concerns regarding potential side effects are essential components of such programs. Additionally, fostering parental autonomy in decision-making regarding vaccination is crucial for enhancing parental intention to vaccinate their daughters against HPV. This study included parents of girls aged between 12 and 15 years. Given that early vaccination against HPV is important, and the individuals aged 9 years or older can receive the HPV vaccination, further studies are needed to include parents of girls and boys aged between 9 and 14 years.

This study has several limitations. First, data collection relied solely on self-reports from parents, which may have introduced single informant bias and social desirability bias. Future research could enhance validity by incorporating diverse data sources. Second, because this was a cross-sectional study, the temporal relationships between the examined variables and parent vaccination intention could not be determined. Third, although multiple factors associated with parental intention were examined in this study, several factors that may be correlated with parental intention to vaccinate their daughters against HPV, such as the perceived threat of HPV infection and related diseases for their daughters, parental HPV-infected status, and the cancer history of parents or other family members were not assessed. Fourth, the study participants were recruited through online advertisements and convenience sampling methods. Logically, the social media used in this study to deliver the message of recruiting participants should cover more than 80% of the cell phone users in Taiwan. However, the response rate of this study was low. This may have something to do with the level of public interest in the study issue. Although this recruitment method can deliver large numbers of participants quickly [41], the sample may not be fully representative of parents of young adolescent girls in Taiwan [42]. Fifth, the total number of girls between the ages of 12 and 15 in Taiwan is approximately 275,000 [43]. Although the sample size of this study was sufficient for regression analysis, further studies that examine the intention to vaccinate daughters against HPV in a population-representative sample of parents are needed. Finally, this study transformed the P-MoVac-COVID19S-12 into the MoVac-HPV-12. However, COVID-19 and HPV infection have different degrees of life-threatening danger. Therefore, the psychometrics of the P-MoVac-HPV-12 warrant further study.

## 5. Conclusions

This study identified several factors significantly associated with parental intention to vaccinate their daughters against HPV, including parents’ perceived values of HPV vaccination for their daughters’ health, concerns regarding the side effects of HPV vaccines, and autonomy in decision-making regarding vaccination. Based on these findings, this study recommends integrating these factors into the design of intervention programs aimed at enhancing parental intention to vaccinate their daughters against HPV.

## Figures and Tables

**Table 1 vaccines-12-00529-t001:** Parents’ and girls’ characteristics.

	*n* (%)	Mean (SD)	Range
**Parents (*n* =** **213)**			
Sex			
Women	172 (80.8)		
Men	41 (19.2)		
Age (year)		46.2 (4.1)	35–60
Education level			
High school or below	51 (23.9)		
College or above	162 (76.1)		
Poor mental health status ^a^			
No	165 (77.5)		
Yes	48 (22.5)		
Parent’s motivation to have daughters vaccinated against HPV ^b^			
Impacts		16.7 (3.3)	3–12
Knowledge		14.7 (3.4)	3–12
Values		16.9 (3.6)	3–12
Autonomy		13.9 (3.4)	4–12
Understanding of the reasons for the government’s promotion of the HPV vaccination		1.9 (0.8)	0–3
Worry about the side effects of the HPV vaccine on their daughters		1.5 (0.7)	0–3
Hearing of side effects after HPV vaccination occurred in some people			
No	128 (60.1)		
Yes	85 (39.9)		
Receiving information of the HPV vaccine from social media			
No	95 (44.6)		
Yes	118 (55.4)		
Intention to vaccinate daughters against HPV		7.4 (2.9)	1–10
**Girls (*n* =** **213)**			
Age (year)		13.8 (1.0)	12–15

^a^: Measured using the 5-Item Brief Symptom Rating Scale. ^b^: Measured using the parent version of the Motors of Human Papillomavirus Vaccination Acceptance Scale. HPV: human papillomavirus.

**Table 2 vaccines-12-00529-t002:** Factors related to parental intention to vaccinate their daughters against HPV: Bivariable linear regression analysis.

	B (SE)	*p*
Parents’ sex ^a^	−0.972 (0.497)	0.049
Parents’ age	0.014 (0.048)	0.773
Parents’ education level ^b^	0.879 (0.459)	0.057
Parents’ poor mental health status	−0.455 (0.472)	0.336
Parents’ perceived side effects of the HPV vaccination on their daughters’ health	0.618 (0.042)	<0.001
Parents’ knowledge of the HPV vaccination	0.519 (0.045)	<0.001
Parents’ perceived value of the HPV vaccination for their daughters’ health	0.626 (0.034)	<0.001
Parents’ autonomy in deciding whether to have their daughters vaccinated against HPV	0.305 (0.055)	<0.001
Parents’ understanding of the government’s rationale for promoting the HPV vaccination	1.043 (0.239)	<0.001
Parents’ concerns regarding the side effects of the HPV vaccine for their daughters	−2.310 (0.223)	<0.001
Parents’ awareness of reported side effects of the HPV vaccination in some individuals	−1.165 (0.396)	0.004
Parents’ exposure to information related to the HPV vaccine from social media	−0.352 (0.397)	0.376
Girls’ age	−0.018 (0.191)	0.924

^a^: Women as the reference. ^b^: High school or below as the reference. HPV: human papillomavirus. SE: standard error.

**Table 3 vaccines-12-00529-t003:** Factors related to the parental intention to vaccinate daughters against HPV: Stepwise multivariable linear regression analysis.

	B (SE)	*p*
Parents’ perceived value of the HPV vaccination for their daughters’ health	0.524 (0.038)	<0.001
Parents’ concerns regarding the potential side effects of the HPV vaccine for their daughters	−0.762 (0.203)	<0.001
Parents’ autonomy in deciding whether to have their daughters vaccinated against HPV	0.086 (0.038)	0.026

Women as the reference. High school or below as the reference. HPV: human papillomavirus. SE: standard error.

## Data Availability

The data are available upon reasonable request to the corresponding authors.

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
