# Peer review of "Determinants of Parental Intention to Vaccinate Young Adolescent Girls against the Human Papillomavirus in Taiwan: An Online Survey Study"

_vaccines, 2024, doi:10.3390/vaccines12050529_

Round 1
Reviewer 1 Report
Comments and Suggestions for Authors
Congratulations to the authors on the quality of the presentation of their study. The manuscript is clear and well detailed. I have added a few comments directly to the PDF to make the article easier for readers to understand.

Author Response
Response to the comments from Reviewer 1
Dear reviewer:
We appreciated your valuable comments. As discussed below, we have revised our manuscript with underlines based on your suggestions. Please let us know if we need to provide anything else regarding this revision.
Comment 1
Add study design into the title.
Response
Thank you for your comment. We added the study design “An Online Survey Study” into the title. Please refer to line 3-4.
Comment 2
Add some numeric results from the regression analysis in the abstract.
Response
Thank you for your comment. We added numeric results from the regression analysis in the abstract. Please refer to line 37-42.
“Parents who perceived greater value in HPV vaccination for their daughters’ health (B = 0.524, standard error [se] = 0.039, p < 0.001) and had greater autonomy in decision-making regarding vaccination (B = 0.086, se = 0.038, p = 0.026) exhibited a higher intention to vaccinate their daughters against HPV. Conversely, parents who expressed greater concern regarding the side effects of HPV vaccines for their daughters had a lower intention to vaccinate (B = –0.762, se = 0.203, p < 0.001).”
Comment 3
Was a sample calculation carried out for these studies, or was it a programmatic sample?
Response
Thank you for your comment. We added the explanation for the method to determine the sample size for this study into Methods section. Please refer to line 185-189.
“According to Green [35], a total number of participants in the studies used a regression analysis model needs at least 50+8*(the number of independent variables). There were 13 independent variables in this study; this study needed at least 154 participants. Therefore, 213 parents were enough for the regression analysis used in this study.
- Green, S.B. How many subjects does it take to do a regression analysis. Multivariate Behav. Res. 1991, 26, 499–510.”
Reviewer 2 Report
Comments and Suggestions for Authors
This is an interesting paper. The only major issue is the author's assessment of normality of data. There are many standard tests for normality, see for example https://www.ncbi.nlm.nih.gov/pmc/articles/PMC6350423/
Author Response
Response to the comments from Reviewer 2
Dear reviewer:
We appreciated your valuable comments. As discussed below, we have revised our manuscript with underlines based on your suggestions. Please let us know if we need to provide anything else regarding this revision.
Comment 1
The only major issue is the author's assessment of normality of data. There are many standard tests for normality, see for example https://www.ncbi.nlm.nih.gov/pmc/articles/PMC6350423/
Mishra P, Pandey CM, Singh U, Gupta A, Sahu C, Keshri A. Descriptive statistics and normality tests for statistical data. Ann Card Anaesth. 2019 Jan-Mar;22(1):67-72. doi: 10.4103/aca.ACA_157_18
Response
Thank you for your suggestion. We adopted the criteria proposed by the references you suggested. Please refer to line 269-272.
“The normal distribution of continuous variables was assessed using criteria of absolute values of <4 and <2 for kurtosis and skewness, respectively [39,40]. These tests indicated that the data were normally distributed.”
- Kim, H.Y. Statistical notes for clinical researchers: Assessing normal distribution (2) using skewness and kurtosis. Restor. Dent. Endod. 2013, 38, 52–54.
- Mishra, P.; Pandey, C.M.; Singh, U.; Gupta, A.; Sahu, C.; Keshri, A. Descriptive statistics and normality tests for statistical data. Ann. Card. Anaesth. 2019, 22, 67–72.
Reviewer 3 Report
Comments and Suggestions for Authors
In the manuscript entitled “Determinants of Parental Intention to Vaccinate Young Adolescent Girls Against Human Papillomavirus in Taiwan” the authors examined the factors that are associated with parental intention to vaccinate their daughters against HPV. The manuscript is well organized and data are sufficiently presented. However, some points need to be addressed.
Authors are required to provide more information concerning the available HPV vaccines (Gardasil; Merck, Cervarix; GlaxoSmithKline, Gardasil 9; HPV nine—valent vaccine), including the date of FDA approval, as well as the HPV types that they protect against. It would be essential to discuss the reasons why HPV vaccines are considered safer than vaccines that contain attenuated or inactivated viruses. Furthermore, it is known that vaccination against HPVs generates prolonged antibody titers and it offers 10- to 100-fold higher titers of L1 specific neutralizing antibodies compared to natural infection. This information would help readers to better understand the efficacy of HPV vaccination. Please provide the appropriate references (Viruses. 2022 Dec 31;15(1):141. doi: 10.3390/v15010141, Expert Rev Vaccines. 2013 Feb;12(2):129-41. doi: 10.1586/erv.12.151, Expert Rev Vaccines. 2010 Oct;9(10):1149-76. doi: 10.1586/erv.10.115. ).
Author Response
Response to the comments from Reviewer 3
Dear reviewer:
We appreciated your valuable comments. As discussed below, we have revised our manuscript with underlines based on your suggestions. Please let us know if we need to provide anything else regarding this revision.
Comment 1
Authors are required to provide more information concerning the available HPV vaccines (Gardasil; Merck, Cervarix; GlaxoSmithKline, Gardasil 9; HPV nine—valent vaccine), including the date of FDA approval, as well as the HPV types that they protect against. It would be essential to discuss the reasons why HPV vaccines are considered safer than vaccines that contain attenuated or inactivated viruses. Furthermore, it is known that vaccination against HPVs generates prolonged antibody titers and it offers 10- to 100-fold higher titers of L1 specific neutralizing antibodies compared to natural infection. This information would help readers to better understand the efficacy of HPV vaccination. Please provide the appropriate references (Viruses. 2022 Dec 31;15(1):141. doi: 10.3390/v15010141, Expert Rev Vaccines. 2013 Feb;12(2):129-41. doi: 10.1586/erv.12.151, Expert Rev Vaccines. 2010 Oct;9(10):1149-76. doi: 10.1586/erv.10.115. ).
Response
Thank you for your comment. We added a paragraph to introduce them and cited related studies. Please refer to line 72-83.
“Currently, the United States Food and Drug Administration (FDA) has approved three prophylactic vaccines for HPV [6]. Gardasil (Merck) is the first vaccine approved by the FDA in 2006 and provides protection against HPV genotypes 6, 11, 16, and 18. Next, Cervarix (GlaxoSmithKline) was approved by the FDA in 2009 and provides immunity against HPV16 and HPV18. The FDA further approved a supplemental use of Gardasil 9 (HPV nine-valent vaccine) in 2014 to expand its protection against HPV 6, 11, 16, 18, 31, 33, 45, 52 and 58 [7–10]. The design of the currently available vaccines is based on virus-like particles (VLPs) that are generated by the major capsid protein L1 and mimics the structure of virions [11–15]. Because that VLPs do not contain viral genome, they are considered safer than attenuated or inactivated viruses that could be turned into infectious ones [8,9]. Vaccination against HPVs generates prolonged, 10- to 100-fold higher titers of L1 specific neutralizing antibodies compared to natural infection [7–9,15]. ”
- Tsakogiannis, D.; Nikolaidis, M.; Zagouri, F.; Zografos, E.; Kottaridi, C.; Kyriakopoulou, Z.; Tzioga, L.; Markoulatos, P.; Amoutzias, G.D.; Bletsa, G. Mutation profile of HPV16 L1 and L2 genes in different geographic areas. Viruses 2022, 15, 141.
- Wang, J.W.; Roden, R.B. Virus-like particles for the prevention of human papillomavirus-associated malignancies. Expert. Rev. Vaccines 2013, 12, 129–141.
- Roldão, A.; Mellado, M.C.; Castilho, L.R.; Carrondo, M.J.; Alves, P.M. Virus-like particles in vaccine development. Expert. Rev. Vaccines 2010, 9, 1149–1176.
- Hirth, J. Disparities in HPV vaccination rates and HPV prevalence in the United States: A review of the literature. Hum. Vaccines Immunother. 2019, 15, 146–155.
- Day, P.M.; Gambhira, R.; Roden, R.B.S.; Lowy, D.R.; Schiller, J.T. Mechanisms of human papillomavirus type 16 neutralization by L2 cross-neutralizing and L1 type-specific antibodies. J. Virol. 2008, 82, 4638–4646.
- The GlaxoSmithKline Vaccine HPV-007 Study Group; Romanowski, B.; de Borba, P.C.; Naud, P.S.; Roteli-Martins, C.M.; De Carvalho, N.S.; Teixeira, J.C.; Aoki, F.; Ramjattan, B.; Shier, R.M.; et al. Sustained efficacy and immunogenicity of the human papillomavirus (HPV)-16/18 AS04-adjuvanted vaccine: Analysis of a randomised placebo-controlled trial up to 6.4 years. Lancet 2009, 374, 1975–1985.
- Harper, D.M.; Franco, E.L.; Wheeler, C.M.; Moscicki, A.-B.; Romanowski, B.; Roteli-Martins, C.M.; Jenkins, D.; Schuind, A.; Costa Clemens, S.A.; Dubin, G.; et al. Sustained efficacy up to 4.5 years of a bivalent L1 virus-like particle vaccine against human papillomavirus types 16 and 18: Follow-up from a randomised control trial. Lancet 2006, 367, 1247–1255.
- Mao, C.; Koutsky, L.A.; Ault, K.A.; Wheeler, C.M.; Brown, D.R.; Wiley, D.J.; Alvarez, F.B.; Bautista, O.M.; Jansen, K.U.; Barr, E. Efficacy of Human Papillomavirus-16 Vaccine to Prevent Cervical Intraepithelial Neoplasia: A randomized controlled trial. Obstet. Gynecol. 2006, 107, 18–27.
Reviewer 4 Report
Comments and Suggestions for Authors
Well written manuscript.
1. It is unclear as to why Taiwan chose to vaccinate girls only. HPV vaccination is not gender specific.
2. It is unclear as to why you have included only 12 to 15 year age group. The age group recommended is 9 years to 14 years for 2 doses and 3 doses for all those beyond
3. What is the actual number of girls that are eligible for vaccination in the age group you have selected - 12 to 15 years in Taiwan?
4. Assuming that each parent who responded represents 1 girl in the age group, what would your sample size be representative of?
5. It is important to realise that mothers who have had a screening test for HPV are more likely to get their daughters vaccinated. Did any of the respondents have a HPV screening done?
6. It is also important to understand that if either parent or another family member has had cancer, the parent is more likely to have their wards vaccinated.
7. You have claimed that you have used several social media platforms to reach out to respondents. Roughly, to how many people did this message reach, and, what percentage of each platform have responded?
8. What would you think were the very low response rate?
9. It is your assumption that the COVID 19 questionnaire is simply validated by substituting covid 19 with HPV. How did you come to such a conclusion? Covid infection is often fatal, immediately, in the un-vaccinated. However, the same is not true of HPV infection. Hence, the questionnaire does not appear to be valid.
Author Response
Response to the comments from Reviewer 4
Dear reviewer:
We appreciated your valuable comments. As discussed below, we have revised our manuscript with underlines based on your suggestions. Please let us know if we need to provide anything else regarding this revision.
Comment 1
It is unclear as to why Taiwan chose to vaccinate girls only. HPV vaccination is not gender specific.
Response
Thank you for your comment. The World Health Organization (WHO) recommends targeting females aged 9–14 years as the primary demographic for HPV vaccination. Accordingly, Taiwan’s Health Promotion Administration subsidize free HPV vaccination for junior high school girls since 2018. Taipei City, the Capital of Taiwan, will also subsidize free Gardasil 9 vaccination for junior high school boys from September 2024 onwards. We described these introductions in Introduction section. Please refer to line 88-97.
“…the World Health Organization (WHO) recommends targeting females aged 9–14 years as the primary demographic for HPV vaccination. Moreover, achieving a vaccination rate of 80% or higher in this age group could mitigate the risk of HPV infection in males [17].
In accordance with WHO’s recommendation, Taiwan’s Health Promotion Administration has been actively promoting the free HPV vaccination service. The first step is to subsidize free HPV vaccination for junior high school girls since 2018 [18]. The goal is to enhance the immunity of young Taiwanese females before they become sexually active, thereby reducing the risk of HPV infection [18]. Taipei City will also subsidize free Gardasil 9 vaccination for junior high school boys from September 2024 onwards [19].”
Comment 2
It is unclear as to why you have included only 12 to 15 year age group. The age group recommended is 9 years to 14 years for 2 doses and 3 doses for all those beyond
Response
Thank you for your comment. Girls aged between 12 and 15 years old are likely to be in junior high school in Taiwan and thus likely to be eligible for free vaccination subsidized by Taiwan’s Health Promotion. Therefore, this study included the parents of girls aged between. We added the explanation into the revised manuscript. Furthermore, we agree that early vaccination against HPV is important; further studies are needed to include parents of girls and boys aged between 9 and 14 years.
“…girls in this age bracket are likely to be in junior high school in Taiwan and thus likely to be eligible for free vaccination subsidized by Taiwan’s Health Promotion.” Please refer to line 170-172.
“The present study included parents of girls aged between 12 an d15 years. Given that early vaccination against HPV is important and the individuals aged 9 years or older can have HPV vaccination, further studies are needed to include parents of girls and boys aged between 9 and 14 years.” Please refer to line 403-406.
Comment 3
What is the actual number of girls that are eligible for vaccination in the age group you have selected - 12 to 15 years in Taiwan?
Response
We added the number of girls aged between 12 and 15 years in Taiwan into the revised manuscript. We added. Please refer to line 423-424.
“The total number of girls between the ages of 12 and 15 in Taiwan is approximately 275,000 [43].”
- National Developmental Council. Population Projections for the R.O.C. (Taiwan). Available at: https://pop-proj.ndc.gov.tw/Custom_Detail_Statistics_Search.aspx?n=39&_Query=cc7bdd82-1915-4943-85c2-95d8cb711265 (accessed on 1 May 2024).
Comment 4, 7 and 8
- Assuming that each parent who responded represents 1 girl in the age group, what would your sample size be representative of?
- You have claimed that you have used several social media platforms to reach out to respondents. Roughly, to how many people did this message reach, and, what percentage of each platform have responded?
- What would you think were the very low response rate?
Response
Thank you for your comment. In the revised manuscript, we made the revisions to address the sample size and sample representation.
- First, we added the explanation for the method to determine the sample size for this study into Methods section.
“According to Green [35], a total number of participants in the studies used a regression analysis model needs at least 50+8*(the number of independent variables). There were 13 independent variables in this study; this study needed at least 154 participants. Therefore, 213 parents were enough for the regression analysis used in this study.” Please refer to line 185-189.
- Green, S.B. How many subjects does it take to do a regression analysis. Multivariate Behav. Res. 1991, 26, 499–510.”
- Second, we added the necessity of further studies examining the intention to vaccinate daughters against HPV in a population-representative sample of parents into Discussion section.
“Fifth, the total number of girls between the ages of 12 and 15 in Taiwan is approximately 275,000 [43]. Although the sample size of this study was sufficient for regression analysis, further studies that examine the intention to vaccinate daughters against HPV in a population-representative sample of parents are needed.” Please refer to line 422-426.
- Third, we also listed the recruitment of participants via social media as one of the limitations in this study.
“Logically, the social media used in this study to deliver the message of recruiting participants should cover more than 80% of the cell phone users in Taiwan. However, the response rate of this study was low. This may have something to do with the level of public interest in the study issue. Although this recruitment method can deliver large numbers of participants quickly [41], the sample may not be fully representative of parents of young adolescent girls in Taiwan [42].” Please refer to line 417-422.
- Bobkowski, P.; Smith, J. Social media divide: Characteristics of emerging adults who do not use social network websites. Media Cult. Soc. 2013, 35, 771–781.
- Whitaker, C.; Stevelink, S.; Fear, N. The use of Facebook in recruiting participants for health research purposes: A systematic review. J. Med. Internet Res. 2017, 19, e290.
Comment
- It is important to realise that mothers who have had a screening test for HPV are more likely to get their daughters vaccinated. Did any of the respondents have a HPV screening done?
- It is also important to understand that if either parent or another family member has had cancer, the parent is more likely to have their wards vaccinated.
Response
Thank you for your comment. We agree that these factors may be correlated with parental intention to vaccinate their daughters against HPV, though this study did not include these factors for examination. We added it as one of the limitations of this study. Please refer to line 411-416.
“Third, although multiple factors associated with parental intention were examined in this study, several factors that may be correlated with parental intention to vaccinate their daughters against HPV such as the perceived threat of HPV infection and related diseases for their daughters, parental HPV-infected status, and the cancer history of parents or other family members were not assessed.”
Comment
- It is your assumption that the COVID 19 questionnaire is simply validated by substituting covid 19 with HPV. How did you come to such a conclusion? Covid infection is often fatal, immediately, in the un-vaccinated. However, the same is not true of HPV infection. Hence, the questionnaire does not appear to be valid.
Response
Thank you for your comment. We agree that COVID-19 and HPV infection have different degrees of life-threatening danger. The psychometrics of the P-MoVac-HPV-12 warrant further study. We added it as one of the issues warranted further study. Please refer to line 426-429.
“Finally, the present study transformed the P-MoVac-COVID19S-12 into the MoVac-HPV-12. However, COVID-19 and HPV infection have different degrees of life-threatening danger. Therefore, the psychometrics of the P-MoVac-HPV-12 warrant further study.”
Reviewer 5 Report
Comments and Suggestions for Authors
My congratulations to the authors on a well conducted and thoroughly reported study. I am attaching a pdf with some minor suggested changes.

Author Response
Response to the comments from Reviewer 5
Dear reviewer:
We appreciated your valuable comments. As discussed below, we have revised our manuscript with underlines based on your suggestions. Please let us know if we need to provide anything else regarding this revision.
Comment 1
Line 131-133: This sentence appears to replicate the previous sentence. Could you just add the citation [19] to the previous statement?
Response
Thank you for your suggestion. Accordingly, we deleted this sentence and added the citation [19] (now changed into [28] in the revise manuscript) to the previous statement. Please refer to line 142.
Comment 2
Line 139-141: citation?
Response
We added the citation (reference [32]) into the revised manuscript. Please refer to line 150.
Comment 3
Line 191: What language used for this and subsequent surveys?
Response
- We used the traditional Chinese version in this study. We added it into the revised manuscript. Please refer to line 206.
- The present study transformed the P-MoVac-COVID19S-12 into the MoVac-HPV-12. However, COVID-19 and HPV infection have different degrees of life-threatening danger. Therefore, the psychometrics of the P-MoVac-HPV-12 warrant further study. We added it as one of the issues warranted further study. Please refer to line 426-429.
“Finally, the present study transformed the P-MoVac-COVID19S-12 into the MoVac-HPV-12. However, COVID-19 and HPV infection have different degrees of life-threatening danger. Therefore, the psychometrics of the P-MoVac-HPV-12 warrant further study.”
Comment 4
Were Japanese media reports disseminated in Taiwan before, during or after your survey?
Response
Japanese media reports were disseminated in Taiwan before our survey. We added it into the revised manuscript. Please refer to line 344.
“…before our survey, the dissemination of Japanese reports on the side effects of HPV vaccination by Taiwanese media have contributed to parental unease”
Round 2
Reviewer 4 Report
Comments and Suggestions for Authors
Thank you for your replies.
The assumption that only girls need vaccination is invalid. HPV causes 6 different cancers: cervix, vagina and vulva in women, penile in men, anal and oropharyngeal in both men and women. Hence, using gender as a crutch is not valid for defending gender discriminating HPV vaccination.
Your sample size constitutes only 0.07745% of the population at risk.